# What, how, when and who of trial results summaries for trial participants: stakeholder-informed guidance from the RECAP project

Hanne Bruhn,[1] Marion Campbell,[1] Vikki Entwistle,[1] Rosemary Humphreys,[2] Sandra Jayacodi,[2] Peter Knapp [3,4] Juliet Tizzard,[5] Katie Gillies [1]

[1]Health Services Research Unit, University of Aberdeen, Aberdeen, UK
[2]Patient Partner, University of Aberdeen, Aberdeen, UK
[3]Health Sciences, University of York, York, UK
[4]Hull York Medical School, University of York, York, UK
[5]Health Research Authority, London, UK

**Correspondence to**
Dr Katie Gillies;
k.gillies@abdn.ac.uk

## ABSTRACT

**Objective** To generate stakeholder informed evidence to support recommendations for trialists to implement the dissemination of results summaries to participants.

**Design** A multiphase mixed-methods triangulation design involving Q-methodology, content analysis, focus groups and a coproduction workshop (the REporting Clinical trial results Appropriately to Participants project).

**Setting** Phase III effectiveness trials.

**Participants** A range of participants were included from ongoing and recently completed trials, public contributors, trialists, sponsors, research funders, regulators, ethics committee members.

**Results** Fewer than half of the existing trial result summaries contained information on the clinical implications of the study results, an item deemed to be of high importance to participants in the Q-methodology study. Priority of inclusion of a thank you message varied depending on whether considering results for individuals or populations. The need for personally responsive modes of sharing trial result summaries was highlighted as important. Ideally, participants should be the first to know of the results with regard to the timing of sharing results summaries but given this can be challenging it is therefore important to manage expectations. In addition to patients, it was identified that it is important to engage with a range of stakeholders when developing trial results summaries.

**Conclusions** Results summaries for trial participants should cover four core questions: (1) What question the trial set out to answer?; (2) What did the trial find?; (3) What effect have the trial results had and how will they change National Health Service/treatment?; and (4) How can I find out more? Trial teams should develop appropriately resourced plans and consult patient partners and trial participants on how 'best' to share key messages with regard to content, mode, and timing. The study findings provide trial teams with clear guidance on the core considerations of the 'what, how, when and who' with regard to sharing results summaries.

## INTRODUCTION

Clinical trials would not be possible without participants. Since 2018 the World Medical Association Declaration of Helsinki has presented the provision of trial results to

### Strengths and limitations of this study

⇒ Focuses on phase III pragmatic effectiveness trials so need to consider that the relative importance of what content is shared with participants may differ for earlier phase trials.
⇒ Research was set within the UK context and as such the legislative and regulatory requirements of trials run elsewhere may vary.
⇒ A significant strength is the coproduction of the guidance with a range of stakeholders who had breadth and depth of trial experience.
⇒ The multicomponent, mixed-methods, design was both progressive (each building on the last) and integrative and a strength of the research.

participants as an ethical requirement.[1] Triallists have sought to become more transparent over recent decades, including via protocol registration, open access publication and enhanced patient and public involvement (PPI); a move to routine results sharing with participants is consistent with this movement.[2]

Most trial participants want to receive a results summary and an audit of the UK Research Ethics application system over 2012–2017 found that most trial teams (87.7%) intended to disseminate results to participants.[3 4] However, these intentions are often not translated into action or not reported as actioned.[4 5] A recent survey of authors of trials indexed in PubMed identified that only 27% reported having disseminated results to participants with a further 13% planning on doing so, however, 33% had no intention of doing so and the intentions of the remaining 25% was unclear.[5] Also, the reporting of whether and how trial results have been shared with participants was not done routinely with 74.9% of final reports not mentioning whether results had been shared with participants.[4] Key among

several reasons for this are researchers' concerns about the potential for results to raise anxiety, and a lack of practical guidance.[5] Some European guidance exists, but it was not coproduced with trial stakeholders and does not provide evidence on the core considerations for production and implementation.[6 7] Little is known about what participants actually want included when results are made available to them.[8] Similarly the World Medical Association (WMA) and UK Health Research Authority (HRA) recommendations for provision of trial results do not guide researchers on content.[1 2] A recent literature review identified a dearth of evidence on how best to share results.[8] Since stakeholder groups may vary in what they think should be shared with trial participants, there is value in coproducing guidance with a diverse group.

The aim of the RECAP (REporting Clinical trial results Appropriately to Participants) study was to generate stakeholder-informed evidence to support recommendations for trialists to implement the dissemination of results summaries to participants.

## METHODS
### Study design
RECAP was a multiphase, mixed-methods study focussing on provision of results summaries for participants of phase III pragmatic effectiveness trials (phases a–d outlined below). Stakeholder identification and recruitment was similar across all RECAP phases (online supplemental table 1). Participants were recruited through organisations that emailed invitation materials on behalf of the study team or posted adverts on social media. All data were collected October 2018–November 2019.

### Q-methodology to identify participant priorities
Q-methodology is a formal method to facilitate the ranking of a set of predefined items.[9] It produces a combination of quantitative (using ranking and factor analysis) and qualitative data (from think aloud interviews) that highlight shared and varying opinions within the population.[9] A 'concourse' or 'Q-set' (in this case of items relevant to reporting of results) is developed and participants place these on a 'Q-sort' grid (see figure 1) to rank their importance.

### Development of the Q set
The concourse was generated through the systematic assessment of publicly available consultations and published guidelines relating to provision of results summaries to trial participants (online supplemental table 2). An initial long list of 239 content items was condensed, through discussion in the project team to identify 28 distinct concepts that formed the Q-set.[9] These content items were presented for a range of different trial scenarios (eg, where the trial demonstrated benefit/ harm of intervention, demonstrated no difference), to establish whether the type of trial results affected the perceived importance of items (online supplemental table 3). Participants were asked to sort for the benefit vignette first and then asked if they would like to move any items after reading subsequent vignettes.

| Least important | | | | Neutral | | | Most important | |
|---|---|---|---|---|---|---|---|---|
| -4 | -3 | -2 | -1 | 0 | +1 | +2 | +3 | +4 |
| Sponsor details | General information about the trial - administrative information | A description of problems encountered/ changes to initial trial plans | Individual results | Issues that may affect the results of the trial | What were the side-effects? | How the trial has contributed to research in the area | Clinical implications of the results | Thank you message |
| | Trial identifier and full title | Declaration of conflict of interests | Statement whether results are applicable to a specific population | Where can I find more information? | Future research - are there any new related or ongoing trials? | If relevant - unblinded information | Topline overview of study results | |
| | | Date this summary was produced | PPI involvement in the trial and its reporting | General information about the trial - scientific information | Treatments being compared | Primary outcome | | |
| | | | Where can I find the full results of the trial? | Future research - are there plans for long-term follow-up in this trial? | Where can I find a more detailed Plain English Summary? | | | |
| | | | A statement that this summary was produced for participants of the trial | Secondary outcomes | Additional information - who can I contact | | | |
| | | | | Characteristics of study population | | | | |

**Figure 1** Viewpoint 1 Q-sort: 'population view'. PPI, patient and public involvement.

## Participant sample

We aimed to purposively sample approximately five participants from a range of stakeholder groups, specifically: PPI partners; members of the public with clinical trial experience; REC members; clinical trial funding body representatives; Sponsor representatives; regulatory representatives; and Clinical Trials Units (CTU) staff/ trialists. Q-methodology you select participants who will potentially reflect a range of views and experiences with overall sample sizes between 12 and 40 people.[10 11]

## Procedure

The Q-sort was conducted in a face-to-face interview at the participant's home or office. Participants were first asked to read the trial scenario, then asked to rank the content items on the Q-sort grid while concurrently explaining why they chose a certain grid position by 'thinking aloud'. Participants were given the opportunity to change item importance for subsequent trial scenarios. Participants were also asked what they thought should happen if the trial participant had died before results were available. The Q-sort grid was photographed after each vignette and/or when a change was made. Interviews were audiorecorded and transcribed verbatim.

## Analysis

Q-sort-specific software was used for analysis (include as Ref: Ken-Q Analysis, A Web Application for Q Methodology V.0.11.1 (15 January 2018). Principal components analysis, most commonly used in Factor Analysis, with varimax rotation (a statistical technique used at one level of a factor analysis as a way to explain the relationship among factors) was applied to identify relationships between individual Q-sorts. In factor analysis, factors are rotated in order to facilitate a more reliable interpretation. A scree plot was investigated for possible factors to be included in the varimax rotation and factors falling around the change in slope before the line levelled off were considered for rotation. Each factor represents a highly intercorrelated group of Q-sorts representing a distinct viewpoint on which content is important in results summaries for trial participants.

Participants' think aloud explanations for ranking statements were organised by factor, Q-set item and vignette to facilitate factor interpretation. Statements for each factor were analysed for overall themes to distinguish between viewpoints being expressed. Analysis focused on the three most and three least highly rated statements in each factor.

## Content analysis of actual trial results summaries

We explored current practice through content analysis of real-life trial results summaries. A request, for examples (focused on pragmatic phase III effectiveness trials of adults with capacity to consent) was disseminated through CTUs on the UK Clinical Research Collaboration registration list,[12] and via the UK Trial Managers Network and social media (eg, Twitter). Trial teams were asked to provide any material (including video or other media) sent to trial participants informing them of results. The analytical framework for the content analysis was based on Q-sort items developed from the HRA guidelines (as detailed above). This framework was systematically applied to all results summaries to ascertain frequency of items. The aggregate results of the content analysis were subsequently compared with the Q-sort findings.

## Focus groups

One meeting was held during which two focus groups of 6–8 trial stakeholders were conducted (by KG and HB), followed by plenary sessions, to explore stakeholders' perception of how and when results should be provided to trial participants. Data collection was guided by semistructured topic guides.

Each group discussed the target topic before they reconvened and shared discussions. Three broad questions were discussed: how and when trial result summaries should be shared; and how do we know when sharing results with participants has been done well. Discussions were audiorecorded and transcribed verbatim. A thematic analytic framework was developed, focusing on the a priori questions about the how and when of sharing trial summaries but leaving scope to identify additional important contributions.[13] This framework was applied systematically to all transcribed text by KG (using NVivo V.12), with coding of a sample checked by HB.

## Coproduction workshop

A coproduction workshop, hosted by the HRA, was held with representatives from each stakeholder group to produce guidance for researchers on developing and providing trial result summaries for their participants. As background knowledge and/or experience of clinical trials was essential to this phase, only patients with trial experience were recruited. Ground rules and facilitation (by HB, KG and JT) sought to ensure all participants had equal voices. The research team presented an overview of the HRA's research transparency agenda (JT) and the evidence generated from all previous phases of the RECAP project (KG). A general discussion of RECAP findings followed before attendees were divided into three smaller mixed-stakeholder groups. The groups considered: what the core content should be for plain language results summaries, what key 'domains' research teams should consider about providing results and who needs to be involved. Iterative rounds of feedback and discussion were conducted after each question with opportunities to raise conflicting opinions but encouragement to reach agreement. Each group made extensive notes during their discussion and key points were reflected and summarised to identify best practice principles.

## Patient and public involvement

Two PPI partners (SJ and RH) were core members of the RECAP Project Advisory Board throughout. RH was involved in setting the research question and contributing

**Table 1** Summary of participants characteristics

| Participant characteristic | Q-sort n=32 | Focus group n=14* | Co-production workshop=21† |
|---|---|---|---|
| Gender | | | |
| Female | 23 (72%) | 7 (50%) | 12 (57%) |
| Male | 9 | 5 | 8 |
| Age | | | |
| Mean (SD) | 52 (14.1) | 52 (16.7) | 56 (12.1) |
| Range in years | 29–77 | 34–81 | 35–75 |
| Country of the UK based in | | | |
| Scotland | 10 | 0 | 2 |
| England | 18 | 11 | 17 |
| Wales | 2 | 1 | 1 |
| Northern Ireland | 2 | 0 | 0 |
| Stakeholder by per protocol group | | | |
| PPI partner | 5‡ | 3‡ | 7‡ |
| Members of the public with clinical trial experience | 5 | 3 | N/A |
| REC members | 7 | 2 | 3 |
| Clinical trial funding body representatives | 4 | 3 | 3 |
| Sponsor representatives | 4 | 1 | 1 |
| Regulatory representatives | 2 | 0 | 1 |
| CTU staff/trialists | 5 | 2 | 6 |

*Only 12 participants completed the demographic questionnaire.
†Only 20 participants completed the demographic questionnaire.
‡Participants may have been identified through funders or regulators but had role of PPI Partner within that organisation.
CTU, Clinical Trials Units; N/A, not available; PPI, patient and public involvement.

to the development of the original grant application. Both were subsequently involved in refining the study design, materials and outputs.

## RESULTS

### Q-sort to determine trial stakeholder views on core content of results summaries for participants

Thirty-two participants completed a Q-sort. 23 (72%) were women, with a mean age of 52 years (range 29–77). The median duration of interviews was 58 min (range 40–105). All participants had some experience of clinical trials and most had experience from several perspectives, for example, as National Health Service (NHS) Research Ethics Committee members and CTU staff or as trial participants and PPI partners (table 1).

A two-factor solution provided the best fit for the data in terms of distinct and interpretable views about what is important to include in results summaries for participants. The two factors together represented 45% of the total explained variance. Participants contributing to each of the factors are presented in online supplemental table 4. Example quotes from the think aloud interviews for the three most and least important items for each factor are presented in table 2.

Factor 1: 'Population view' - what populations would wish to know in trial results summaries and is considered at the level of the trial as a whole (figure 1, table 2)

The Q-sorts of 19 participants loaded onto factor 1. Most were from non-patient participants, namely funders, REC members, Sponsor representatives and trialists. One PPI partner (identified through a funder) contributed to this factor, in which a thank you message was identified as the most important content item to be included in result summaries. Some professionals explained it was important to say thank you as they cannot give anything material to reciprocate the altruistic act of trial participation. The second most important content item in factor 1 was 'Clinical implications of the results'. Participants described its importance in terms of the study impact on clinical practice. The third most important content item in factor 1 was 'Topline overview of study results', which was thought to make the overall results more accessible to participants who do not want details.

Factor 2: 'Individual view'—what individual participants would wish to know in trial results summaries and consider more personally relevant (figure 2, table 2).

The Q-sorts of 13 participants loaded onto factor 2. Most were trial participants and PPI partners, including two professionals (one sponsor representative and one

**Table 2** Quotes from participants during Q-sort think aloud

| **Viewpoint 1—population view** | |
|---|---|
| More important | |
| 1. Thank you message | "It's most important because contrary to belief people give time, and effort into this, and it means something to them to be involved in a study." (ID 13, REC)<br>"I actually think, just saying thank you, even if you don't get anything else out there, just saying thank you is so important to participants.(…)" (ID 16, Sponsor) |
| 2. Clinical implications of the results | "I think that's very important, clinical implications of the results if it's been a real impactful study. I think that would be really good to know. I think individuals would appreciate that. Equally if it hadn't been impactful!" (ID 28, funder) |
| 3. Topline overview of study results | " I'd say that's pretty important because it's the—if you can do it in a top-line overview, it's a lot easier for people to understand than pages and pages of feedback." (ID 12, Trialist) |
| Less important | |
| 1. Sponsor details | "Name and contact of the sponsor. They should have had that from day one, it should be on their information sheet, it's important that they have that. But I would say, at the end of the study feedback, that shouldn't be given to them then because they should have already had it. " (ID 12, trialist—'Population view') |
| 2. General information about the trials—administrative information | "General information about the trial, where, when, start and end dates". Again, I think that's less important than what the results are, how they're going to affect me, side effects, what treatment is better than what, yeah…" (ID 15, REC member) |
| 3. Trial identifier and full title | *"They don't need things like the registration and all of that. I mean every letter they receive should have the trial title on the top, so why would they need… so I'm not giving that much priority.*" (ID 31, Regulatory Representative—'Population view') |
| **Viewpoint 2—individual view** | |
| More important | |
| 1. The primary outcome | "Because they are the primary outcomes! Sorry, that's a wishy-washy answer. But because I'm assuming that is what the research was done for was for the primary outcomes, to see what the whole point of the research was for. Without the primary outcomes you might as well not bother with the research itself. I will want to know, when reading something, what the outcome was, otherwise why bother reading the document?" (ID 26, Member of public with trial experience) |
| 2. Clinical implications of the results | "(…)This is what it's about, because if it doesn't filter down to the doctors or the NHS, then what's the point." (ID 5, PPI (funder)) |
| 3. What were the side effects? | "I think "Side effects", I mean, you will know as a participant what your personal experience of side effects are, but I think it's really good to know what the population as a whole experienced." (ID 8, regulatory representative)<br>"Well I suppose it would depend on which side of the trial you were on. You might know what the side effects are already but you might not, and I suppose is that worth… is the benefit outweighed by the side effects or is the side effects outweighed by the benefits, what would be more important to somebody? And I suppose that's to do with quality of life." (ID 21, Member of public with trial experience) |
| Less important | |
| 1. Thank you message | "I'm probably really weird but I'm really not bothered if I get thanked for doing stuff if it's something that I want to do, you know, like internally motivated to do it. It just wouldn't make any difference. It's nice to have a friendly member of staff more than somebody saying thank you all the time. I'd rather just be treated well when I was there, I think, so that's definitely quite a low one." (ID 3, PPI)<br>"Well to me it's your moral duty to do it [take part in a trial] and it's something that do you know, it's like well if somebody says, "Oh, thanks very much" that's fine, but to actually go to the time and trouble to put a thank you message out… if it's at the bottom of a letter fine, but if it's a separate, "We'd like to thank you for taking part in this", and it costs postage, it's costs… waste of time, don't bother, that's the way I look at it." (ID 29, Trial participant) |
| 2. Trial identifier and full title | "Trial identifier and full title … again, these seem like administrative things that are not really telling me what I want to know." (ID 5, PPI—'Individual view') |
| 3. Sponsor details | Yeah, I mean that [Sponsor details] would be interesting, but I think I'd probably know that from the beginning, so." (ID 30, PPI—'Individual view') |

NHS, National Health Service.

| -4 | -3 | -2 | -1 | 0 | 1 | 2 | 3 | 4 |
|---|---|---|---|---|---|---|---|---|
| Thank you message | Trial identifier and full title | Date this summary was produced | Where can I find a more detailed Plain English Summary? | Treatments being compared | How the trial has contributed to research in the area | Issues that may affect the results of the trial | Clinical implications of the results | Primary outcome |
| | Sponsor details | Where can I find the full results of the trial? | Declaration of conflict of interests | Topline overview of study results | Statement whether results are applicable to a specific population | Individual results | What were the side-effects? | |
| | | A statement that this summary was produced for participants of the trial | PPI involvement in the trial and its reporting | Future research - are there plans for long-term follow-up in this trial? | If relevant - unblinded information | Secondary outcomes | | |
| | | | Where can I find more information? | Future research - are there any new related or ongoing trials? | A description of problems encountered/ changes to initial trial plans | | | |
| | | | General information about the trial - administrative Information | Characteristics of study population | General information about the trial - scientific information | | | |
| | | | | Additional information - who can I contact | | | | |

**Figure 2** Viewpoint 2 Q-sort: 'individual view'. PPI, patient and public involvement.

regulator representative) and one (lay) REC member. In this factor 'the primary outcome' was the most important content item, typically because it provides an answer to the main trial research question. The second most important content item as in factor 1, was 'clinical implications of the results'. Participants loading onto this factor emphasised that the findings needed to be used to change treatment in the NHS (if appropriate). The third most important content item in factor 2 was 'what were the side effects'. There was a desire for trial participants to be able to compare their own experiences of possible side effects to that of everyone else in the trial. Some RECAP participants also expressed a need to judge for themselves whether treatment benefit outweighed negative side effects.

In contrast to factor 1 where 'a thank you message' was ranked the most important content item, it was ranked least important in viewpoint 2. For one participant the thank you at the end is less important than their personal reasons for taking part and the treatment throughout the trial. Others reported trial participation as an act for the common good; not only did they not need formal thanks, but its cost was seen as unnecessary.

There appeared to be consensus across both factors that the 'Sponsor details' and 'Trial identifier and full title' were two of the three least important items; participants stated that this information would have been made available at the beginning of the trial.

### Q-sort changes based on trial context
Two participants considered 'Clinical implications of the results' to be less important when considering the vignette in which the intervention was found to be less effective than the control. About half did not make any changes between each vignette and those that did were modest (ie, not changing valence).

### Provision of trial results summaries to next of kin following death
Most participants indicated they thought next of kin should have the option of receiving the results summary. One participant, however, did not want their partner to receive the trial results anticipating it would be a painful reminder of her death, and suggesting that this should be considered during the trial consent process. Questions were raised around how much background information next of kin would have about the trial and a potential need to highlight the contribution of the individual and impact of the trial in results summaries to next of kin.

### Content analysis of existing trial results summaries for participants
Sixty-nine trial results summaries were received and 30 were eligible for analysis. Reasons for ineligibility largely related to summaries being from trials involving children. The 30 eligible summaries were provided from 10 host organisations. They included paper and video-based reports from academic and industry-sponsored trials involving a range of clinical areas and interventions.

All 30 summaries (100%) reported the 'primary outcome' of the trial and all but one (n=29, 97%) also reported the' treatments being compared'. The third most frequently reported item was a 'thank-you message', reported in 26 (87%) summaries. Three items were not included in any summary: 'declarations of conflict of interests'; 'issues that may affect the results of the trial'; and 'individual results'. Additional items that were not captured by the original framework

included information about the funder (n=11) and the trial cost (n=9). See online supplemental table 5 for full results.

Comparison of Q-sort findings to content analysis of existing trial results summaries: The most and least frequently identified items from the content analysis of existing trial summaries were compared with the items in the Q set. First, comparison was made of the three most and least important items identified in both the individual and population viewpoints against the frequency of reporting in existing trial result summaries (table 3A). Overall this highlighted that several items deemed relatively unimportant by Q-sort stakeholders are routinely reported in existing results summaries (ie, Trial identifier and full title, General information about the trial, Thank you message). The 'Clinical implications' item, which was viewed as important in both Q-sort viewpoints was only reported in 43% of trial results summaries analysed. Table 3B presents the comparison of the three most and least frequently reported items identified in the content analysis of results summaries compared with the scoring within each viewpoint on the Q-sort. Overall there appeared to be similarities on the frequency of items in existing summaries and their relative importance in the viewpoints. However for two items identified as being important by the individual viewpoint, 'Issues that may affect the results of the trial' and 'Individual results' this was not the case: they were not identified in any of the summaries analysed. Full results of the comparison of the Q-sort and content analysis can be seen table 3.

### Focus groups to explore the how and when to provide trial results summaries to participants

Fourteen participants contributed to two focus groups (see table 1 for participant demographics).

When asked how and when trial result summaries should be provided to trial participants, stakeholders identified a range of considerations that trial teams should take on board. These largely related to communication planning, with concern to ensure summaries were contextually relevant and participant-focused, and to foster equitable partnerships between participants and trial teams. Key findings are discussed below. Further exemplar quotes are presented in online supplemental table 6).

### Experiences of current practice when sharing trial results summaries with participants

Throughout the discussions, many stakeholders indicated that current processes of sharing results with trial participants are sub-par, compared with other aspects such as informed consent, largely because results sharing may not be a priority for trial teams. Focus group participants who had previously been trial participants reported not receiving results.

> 'No, absolute silence about the whole thing. Absolutely no communication of any sort…'

### What methods should trial teams use to share results with trial participants?

Focus group participants recognised there will not be a one-size fits all approach to sharing results summaries. They recognised that having a plan in place at the beginning was a good first step, and that there should be scope to adapt this responsively

> '…things change, like some trials are five or ten years and you would decide to present the results differently after ten years than you would if they're you know, at the beginning. So I think that although it needs to be planned and considered at the beginning, I think it shouldn't be set in stone.'

Stakeholders agreed that a flexible approach was critical, such that participants could have the opportunity to choose how they want to be informed and providing opportunities for layering of information according to individual preferences. The pros and cons of various methods for delivery were discussed, including whether peer support could be a mechanism through which results could be shared. Participants recognised that what they envisaged as ideal (eg, face-to-face with a person who you have established a relationship with) may not always be possible. The need to consider the demographic or contextual factors of the trial population (eg, trials in older adults or those with a high mortality rate) was discussed, as was communicating results that may disappoint trial participants (eg, a trial that showed no effect, or when reporting results to individuals who received the treatment that was less effective).

> …flexibility, the layered approach to the information so that you can access it in a format that's useful and helpful for you and that they should be as inclusive as possible those formats, tailored to the population.

> …by having the teleconference, people actually had a chance to sit and think about how they felt and reflect on that before they had to then interact with anybody else, so they got to hear the news in the privacy of their own environments.

### When should results summaries be shared with trial participants?

Various considerations were raised about the timing of provision of results summaries. Mirroring the discussions on how to share, it was generally thought important to give participants a choice about when or indeed whether they receive results, and to recognise their views might change over the duration of the trial and be dependent on personal and trial context.

> I think my opinion would have changed during the trial to wanting to know more about it. And it did, I wanted to know whether I'd been on it or what I'd experienced was just a placebo effect. I think I would have started not really knowing what I wanted to know, but as I went on, I definitely wanted to know more.

**Table 3** Comparison of content analysis and Q-sort findings

**A**

Three most and least important items in both Q-sort viewpoints compared with content analysis of existing results summaries

| Item | Trial identifier and full title | Topline overview of study results | Sponsor details | General information about the trial - administrative information | What were the side effects? | Primary outcome | Thank you message | Clinical implications of the results |
|---|---|---|---|---|---|---|---|---|
| % of results summaries reporting item | | | | | | | | |
| Population | 40% | 27% | 27% | 57% | | | 87% | 43% |
| Individual | 40% | 27% | | | 47% | 100% | 87% | 43% |

**B**

Three most and least frequent items identified in content analysis of results summaries compared with scoring on Q-sort

| Item | Declaration of conflict of interests | Treatments being compared | Primary outcome | Issues that may affect the results of the trial | Individual results | Thank you message |
|---|---|---|---|---|---|---|
| Frequency of reporting in results summaries | 0% | 97% | 100% | 0% | 0% | 87% |
| Population | -2 | 1 | 2 | 0 | -1 | 4 |
| Individual | -1 | 0 | 4 | 2 | 2 | -4 |

| | |
|---|---|
| | Denotes more important Q-sort rating |
| | Denotes middle Q-sort rating |
| | Denote less important Q-sort rating |

*Grading depicts strength of importance with deeper colour representing stronger rating.

Several stakeholders voiced that trial participants should be the first to know the results, and ideally before the study is published, but it was recognised that there are challenges associated with this such as embargoes from funders. As a minimum, trial participants should be informed before trial results are more widely publicised.

> a friend of mine who took part in a parenting project with an autistic child and she was in a control group, she knew. And the first she knew of the results was when she read it in the daily paper in one of the papers, which said, 'super parenting aid for autism'. So, you can imagine how that made her feel and she hadn't had any results.

Focus group participants also recognised the importance of managing expectations and balancing a desire to present results to participants as quickly as possible against the time required for scientifically rigorous analysis. A proportionate approach may be appropriate for some trials and informing participants that you will let them know at 'the earliest opportunity' was felt to be suitable.

> We don't want to impair the quality or the rigour of which it's done, and it's got to be lined quite slowly. It's just the balance of that. What we don't want to do is to create levels of expectation among participants that we're going to learn lots too early, because that can be very, very dangerous and disappointing because you're disappointing people.

### How do we know when sharing of result summaries has been done well?

When considering how to determine whether results sharing had been done well, focus group participants proposed a range of options. Many covered positive aspects such as message clarity, tell you what you needed to know/found interesting, trial expectations being met, need for further information, feeling valued and/or recognised, positive experience, future participation, and recommending participation to others (family/friends). However, negative indicators were also identified, including regret about decision (about trial participation) anxiety, and the psychological impact of learning that a trialled treatment had no effect.

> 'Did it meet our expectations?' So, one of the things [focus group participant] said so well, so eloquently, about, if it is going to be a partnership there should be a clear understanding about what's expected on both sides. So, you know, our patient participants are pitching up and giving so much time, so very much time, and therefore they should be able to hold us, as researchers, accountable for meeting our bit of the deal.

### Coproduction workshop to develop practice recommendations for disseminating trial results summaries to participants

Twenty-one people contributed to the final co-production workshop to agree the core content template (table 1) to inform a Plain Language Summaries and broad recommendations for sharing trial results summaries with participants (see figure 3). The core content recommendations present key topic areas with explanations of what content or descriptions would be expected to be provided in plain English summaries including: primary and secondary outcomes; side effects; clinical implications; contribution to research area; issues that may affect the results; applicability to specific populations; and any changes to the original trial. It is important to recognise that this was not produced as a prescriptive list but rather suggestions based on diverse stakeholders considered reflections on experiences to date. Broader good practice recommendations to support appropriate reporting of clinical trials results summaries to participants, including who should be involved, were also developed (see box 1). Overall, these co-produced recommendations highlighted stakeholders' views for the need for early planning when considering sharing trial results summaries with participants such that activity is considered from the start and throughout the trial. It was agreed that this should include consideration of who the recipients are (ie, adults, children, next of kin) which may in turn require different versions. The group also identified the Importance of considering who else needs to be involved in the process for example, patients organisations, clinicians, etc and ensure they are also committed to the activity. Discussion on the mode of delivery for the result summaries concluded it should be decided in collaboration with trial participants and ideally based on personal preferences. Whether and how results summaries will be shared with trial participants who withdraw from the trial was also identified as important by the group and as requiring attention. Stakeholders stated that trial teams should also consider when results will be shared and make this clear to participants to set expectations accordingly. If results are not to be shared (due to politically/culturally sensitive issues), this should also be made clear to participants from the outset. When considering who beyond the trial team should have a role in sharing trial summaries with participants, the co-production group highlighted that funders should provide financial resource and ask for its inclusion in grant applications, regulatory bodies could provide a repository of good practice to provide exemplars, and employers should source training on writing lay summaries.

### DISCUSSION

The RECAP study has coproduced stakeholder-informed recommendations for trialists to implement the dissemination of results summaries to participants. This multiphase multistakeholder study has produced evidence on current practice, defined core content of results

The template below contains the key topic areas for inclusion in the Plain Language Summary of trial results provided to trial participants at the end of a trial. These items are not a prescriptive list but do provide suggestions for good practice that have been co-produced and based on empirical research findings. This list could be supplemented with other relevant items as required by individual trials through development with patient and/or public partners.

Overall the Plain Language Summary should aim to address the following over-arching questions:

- What question the trial set out to answer?
- What did the trial find?
- What effect have the trial results had and how should they change NHS/treatment?
- How can I find out more?

Inclusion of unblinded information or individual level results may not be appropriate for the Plain Language Summary submitted to the HRA but this type of information could be considered by trial teams when providing summary of results to participants. Trial teams need to make it clear in the summary to the HRA whether this unblinded information for individuals was included.

Acknowledgement of the contribution of trial participants should also be made clear in the Plain Language Summary through a general thank you statement to those who have contributed their time as participants in the trial and without whom the trial would not be possible. In addition, it is suggested that trial teams should involve patients and public, for whom the results will be relevant, in the development of the materials to make sure the Plain Language Summaries are understandable.

Key topic areas to include and the content and/or a description for each are provided are below.

---

**Primary outcome**
These are the measures the trial teams decided beforehand would be the most important in deciding how effective the treatments may be and what harm they may cause

**Secondary outcome(s)**
These outcomes are measured in trials of treatment effects that are pre-specified in the protocol as being relevant, but less important than the primary outcomes.

**What were the side-effects?**
A description of any undesired actions or effects of a drug or treatment including how often they occurred and how bad they were.

**Clinical implications of the results**
The clinical implications of the trial results are about what health and social care (or the doctors in the NHS) could do with the results and what it means for them when deciding on a treatment.
If published in a scientific journal provide the reference. If not, state it will be submitted to peer reviewed journal
Clinical implications may need to be placed in context, highlighting that it may take a very long time for results to influence practice.

**How the trial has contributed to research in the area**
This should include a general comment on what this trial contributed to the relevant area of research and any potential next steps to build on that knowledge.
How findings from a trial might help researchers
 - to seek approval for using the treatment for a particular disease
 - in other studies to learn whether these patients are helped by this treatment
- in other studies to compare this treatment with other treatments for patients with this condition/disease

**Issues that may affect the results of the trial**
These include information about an issue that may affect the results of a trial.

**Statement whether results are applicable to a specific population**
If relevant - A statement describing that because the trial was carried out in a specific group of people (e.g. men only or women only groups) it cannot be assumed that it will work or will not work in other groups of people.

**A description of problems /changes to initial trial plans**
Things do not always go to plan and this happens in trials too. This would be a description of any problems that occurred and any changes that had to be made to the plan for the trial to deal with these problems.

---

**Figure 3** RECAP: plain language summary template. HRA, Health Research Authority; NHS, National Health Service; RECAP, REporting Clinical trial results Appropriately to Participants.

summaries and produced a template for trial teams. Furthermore, it considered the critical components of 'how and when' of results sharing and synthesised this information to inform the co-production of a set of actionable recommendations.

There is room for improvement on current practice in terms of the content of trial result summaries. Our study found that fewer than half the examples analysed included information on the clinical implications of the results, an item deemed important by both the individual and population viewpoints in the Q-methodology study. This finding may not be surprising given that the problem also exists for the scientific reporting of trial results (eg, a review of dementia drug trials found less than half discussed clinical

**Box 1    Key considerations to support appropriate reporting of clinical trials results summaries to participants**

⇒ Essential to involve patient partners in the development of the lay summary.
⇒ Ensure that dissemination of results is considered from the start and throughout the trial—a proactive dissemination plan should be developed to sit alongside applications (to funder and research ethics committees) and protocols. Dissemination should be written into grant applications to include appropriate funding (that might cover trial team members to deliver and also external expert to design) to enable the process to happen and be ring fenced accordingly. Information about dissemination of results should be provided within patient information leaflets used to support the informed consent process. This might not be prescriptive (as details would be developed with patient partners during the trial) but it should state when and how they would be available.
⇒ Consider from the outset who the audience is i.e. who are the recipients of the trial result summaries and how might the information and messages need to be tailored to ensure they are understood and interpreted in the correct way (e.g. adults lacking capacity, children, etc). Teams may need to consider different versions for different audiences and the depth and granularity of findings may also need to be adjusted to consider the interest of the audience.
⇒ If results are not to be shared (due to politically/culturally sensitive issues) this should be justified in advance to participants.
⇒ Research teams also need to consider, from the outset, how they will deal with participants who have withdrawn from the trial.
⇒ With regard to the mode of delivery of the results, this should be developed in consultation with patient partners and where possible individuals asked for their preferences to ensure accessibility. Possible options include: email; post; web link; face to face; telephone. The message of the results may also shape decisions about mode of delivery e.g. those with high mortality rates feeding back (to parents/carers/next of kin) may require different modes of delivery to other 'lower risk' trials.
⇒ Consideration of the timing of results i.e. before or after publication is important, and how this will be operationalised.
⇒ Funders and regulators may consider a repository of good practice. This could be an online open access resource that would contain examples of plain English summaries that have been done well and could be used as exemplars by others.
⇒ Research teams should consider who needs to be involved (from both within the immediate team and any external partners/experts) at each stage in terms of: resource; design; implementation and compliance.
⇒ As a minimum research teams should consider training for researchers in writing lay summaries—look to other industries that do this well or other departments within host institution.
⇒ Funders should provide financial support for the dissemination of the trial results and therefore should request a lay summary (which has been developed for trial participants) is provided at the end of the study with any final report. Researchers should also explore opportunities with the communications teams within funders to develop the lay summaries for dissemination.
⇒ Additional partners that may want to be included or consulted in the dissemination process include (but are not limited to): o Patient expert groups o Graphic designers, graphic illustrators, animators
⇒ Funders.
⇒ Journals.
⇒ Scientific writers.

Continued

**Box 1    Continued**

⇒ Royal colleges.
⇒ Clinical commissioning groups.
⇒ Political advisors.

significance of results[14]). Other items that should be considered based on our analysis of current practice and findings from the Q-methodology study are: issues that may affect the trial results, and individual results. Trial teams need to consider whether to provide individual results in addition to overall summaries. Individual results were viewed as important in RECAP yet most summaries we reviewed, and those in the existing literature, have not included them.[8] The results of the Q-methodology study also highlighted a divergence in opinion between the two viewpoints, where the population viewpoint viewed a thank you message as the most important item but the individual viewpoint perceived it as least important. This variability is mirrored in other studies.[15 16] Overall, the Q-methodology study highlights that patients' and more general public preferences for information contained within trial results summaries are likely to differ from other trial stakeholders, further underpinning the need to involve patients and members of public groups during the development of the information.

A further consideration about developing trial results summaries relates to sociocultural influences on desires for and views about the presentation of trial result summaries. With regard to content, for some trials and populations there may be a requirement to consider whether some results are sensitive to characteristics such as sexuality or religion and ensure these are accounted for during development. When considering how to share results, it may be more or less appropriate to provide results online for some communities, and it may sometimes be appropriate to involve respected community leaders, as well as members of the trial team, in the communication and discussion of trial findings. Further research to explore the most appropriate ways to consider the intersections between these characteristics is needed given most research to date has not considered a wide range of diverse experiences.[8]

Our linked scoping review identified that most studies investigating sharing results with participants explored paper-based postal methods, with a few face-to-face and online methods identified.[8] Within the review of current practice, we identified a range of methods used to disseminate trial results. Focus group participants recognised the need for flexibility depending on the features of the trial and results as well as individual needs and preferences, which may change over the trial. It is likely that several other mechanisms and methods of sharing result summaries with trial participants are used in practice but often not shared widely or even within the trials community. Trial teams could share examples of how trial results have been communicated with trial participants to help

inform others teams decision making. A recent good practice example of this involved sharing results of a trial through an online meeting (enforced by the pandemic but allowing a more responsive way to share results). The report of this activity also included the associated costs, which will again help teams plan for such activities in future.[17]

The other main consideration for trial teams is timing: when to provide results summaries. Previous studies have tended to report that summaries were provided at or after publication of results.[8] Participants in the RECAP focus groups voiced that a best-case scenario would be for participants to be the first to receive the results but recognised the challenges around data quality assurance publication embargos. There are specific trial contexts in which these considerations may be even more important. For example, an interview study of parents of surviving babies in a neonatal critical care trial identified a delicate balance between the emotionally charged content of the information with defining an endpoint to difficult events and acknowledging participants contributions to the research endeavour.[18] This study also highlighted that even in challenging contexts, trial participants and their families may be responsive to receiving the results of the trial they contributed to and trial teams should endeavour to provide that information in emotionally sensitive ways. As per the recommendations developed by RECAP, we would propose that (where appropriate) participants are invited to consent in advance for the sharing of the results with next of kin, and that next of kin should have the option of receiving the results as opposed to assuming they desire to receive them.

### Strengths and weaknesses of the study

One of the limitations of the RECAP guidance is that it focused on phase III pragmatic effectiveness trials. While many of the findings and recommendations are likely transferable to other trials, the relative importance of what content is shared with participants may differ for earlier phase trials. It is also worth considering the UK context of RECAP and that the legislative and regulatory requirements of trials run elsewhere may vary. A significant strength is the co-production of the guidance with a range of stakeholders who had breadth and depth of trial experience. Yet more could have been done to include a wider range of participants with lived trial experience as participants or family members of those participants. However, the inclusion of a multiperspective broad expertise base fed into the RECAP study and ensured a diverse range of perspectives influenced the recommendations. The multicomponent design, which was both progressive (each building on the last) and integrative, is a further strength.

### Implications for practice and research

The next steps for dissemination of trial results summaries is to ensure that the process is enacted in accordance with the good practice principles and recommendations RECAP developed. Given the call from stakeholders for clinical implications to be present in the trial result summaries, and a continued push to ensure clinical trials translate into patient benefit, this item should be considered as a core item of trial result summaries.[19] RECAP findings highlight a critical need to ensure trial teams engage early with patient partners and participants to ensure the information they wish to receive is shared accordingly and that expectations are managed appropriately. While there was divergence in the priority of providing a thank you message, given it is unlikely to cause harm and was not deemed unimportant but rather less important compared with other items, inclusion of a thank you message should be considered. A repository of good practice that demonstrates a range of ways in which trial result summaries have been shared would be useful to provide trial teams with exemplars and inspiration. The question of what media to use to share results will require input from participants and may require responsive modes to meet diverse individual needs and preferences. Ensuring participant expectations are met at the informed consent stage with regard to when (and how) they can expect to receive, or request, trial result summaries should be built into the trial planning.

As recognised by this work, and others, these improvements in practice will depend not just on trial teams but also on regulators, funders and journals to consider how to ensure they come about. Preliminary steps such as: the BMJ requirement for authors to describe plans for sharing findings with participants and other relevant communities (or declare they have none); and the HRA plans to add lay summaries of research findings to individual approved studies on their website and use the ethics approval process to ask when and how participants will be informed of results rather than if, are encouraging.

Further research is required to generate evidence on how best to share trial results summaries with participants, for example, through embedded evaluations (Studies Within A Trial) of interventions. In-depth exploratory studies of how best to share results in different contexts also have a place to provide more in-depth understanding of diverse participant experiences of receiving results. Finally, gaining an understanding, both in practice and in research, of whether the sharing of trial results is also being done well is key. Metrics on the number of trials producing Plain Language Summaries will provide useful data, but we also need to understand whether these are achieving desired objectives – which will be less easy to quantify or assess routinely.

### CONCLUSION

The RECAP guidance provides stakeholder-informed recommendations to facilitate how best to disseminate trial results to those who participated. Our research suggests that results summaries for trial participants should cover four core questions: (1) What question did the trial set out to answer?; (2) What did the trial find?;

(3) What effect have the trial results had or how should they change NHS/treatment? and (4) How can I find out more? Trial teams should develop implementable plans, resourced accordingly, for the sharing of trial results summaries with trial participants. Patient partners and trial participants should be consulted on how 'best' to share key messages. The RECAP findings are a key first step in realising this ambition by providing trial teams with clear guidance on the core considerations of the 'what, how, when and who' of sharing results summaries.

**Acknowledgements** We thank all participants for their time and contribution to this study, both those who contributed directly in the Q-sort, focus groups and coproduction workshop and indirectly through sharing of results summaries. Thanks also goes to two trial managers in CHaRT who facilitated recruitment of participants from their ongoing trials, Ms Karen Innes and Dr Gordon Fernie. We are also grateful to Amanda Hunn, Bill Davidson and Jim Elliott for their advice at key project stages. Thanks also to Dr Lorna Aucott for Q-factor analysis advice.

**Contributors** MC, VE, KG, RH and PK had the original idea for the project and contributed to the funding application to the Academy of Medical Sciences Springboard (awarded to KG). MC, VE, KG, RH, PK and SJ were involved as collaborators and contributed to the methodology of the project . HB, KG and JT were involved in data collection. HB and KG conducted the analyses. KG wrote the original draft of the manuscript. All authors contributed to revising and approving the final manuscript. KG acts as the guarantor and accepts full responsibility for the work and/or the conduct of the study, had access to the data, and controlled the decision to publish.

**Funding** RECAP was funded by the Academy of Medical Sciences (SBF002\1014) and KG was funded by a Medical Research Council Strategic Skills Methodology Fellowship (MR/L01193X/1). The Health Services Research Unit, Institute of Applied Health Sciences (University of Aberdeen), is core funded by the Chief Scientist Office of the Scottish Government Health and Social Care Directorates (CZU/3/3).

**Disclaimer** The funders had no involvement in study design, collection, analysis and interpretation of data, reporting or the decision to publish.

**Competing interests** None declared.

**Patient and public involvement** Patients and/or the public were involved in the design, or conduct, or reporting, or dissemination plans of this research. Refer to the Methods section for further details.

**Patient consent for publication** Consent obtained directly from patient(s)

**Ethics approval** This study involves human participants and was approved by NHS London Brent Research Ethics Committee Reference 18/LO/0634. Participants gave informed consent to participate in the study before taking part.

**Provenance and peer review** Not commissioned; externally peer reviewed.

**Data availability statement** All data relevant to the study are included in the article or uploaded as online supplemental information. Not applicable.

**ORCID iDs**
Peter Knapp http://orcid.org/0000-0001-5904-8699
Katie Gillies http://orcid.org/0000-0001-7890-2854

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
