## [Reviewer comments · BMJ Open]

ARTICLE DETAILS

TITLE (PROVISIONAL)	The what, how, when and who of trial results summaries for trial participants: Stakeholder informed guidance from the RECAP project.
AUTHORS	Bruhn, Hanne; Campbell, Marion; Entwistle, Vikki; Humphreys, Rosemary; Jayacodi, Sandra; Knapp, Peter; Tizzard, Juliet; Gillies, Katie

VERSION 1 – REVIEW

REVIEWER	Nair, Satish Tawam Hospital, Acad affairs
REVIEW RETURNED	28-Oct-2021

GENERAL COMMENTS	The Declaration of Helsinki has laid down ethical responsibilities related to clinical trials, and these are restated by the clinical trial regulations in the US, UK, Asia, and the European Union. The guidelines mandate summary results to be provided to study participants in understandable language. Organizing and disseminating trial results to participants is an ethical responsibility. In addition, to the regulatory framework compliance, dissemination has other benefits, aiding in enhancing public trust in clinical research and promoting active participation in future studies. Policy makers and regulators are benefited by easy assimilation of study results in order to generate policies and monitor practices. Regardless of the regulations, three of the key deterrents for the dissemination of trial results have been the concerns of raising public anxiety, and subsequently the fear of legal implications, and loss of public trust. The content, mode, quality, readability, understandability, and selectivity (whether something applies to a subgroup or the whole) of dissemination of clinical trial results still rest in the grey domain of clarity. This is an important study. Given the complexity, the authors have rightly employed a multi-phase mixed-methods triangulation study design using multiple stake holders. The Q methodology is quite appropriate for the study design. 1) The paucity of patient participants from concluded clinical trials is a significant shortcoming of the study design. Irrespective of the feedback of the sponsors, public involvement partners. it is the patient and their family reported outcomes that will enable a superior dissemination design. The authors are urged to explain why patient and their families were not abundantly included in the study design? 2) Q-sort to determine trial stakeholder views on core content of results summaries is another area of concern. The NHS Research, Ethics Committee members, CTU staff, and PPI partners would have appropriate for the design of the structure of the content,
---

	rather than the content itself. The importance of having a patient/patient family led Q-sort is advised to design the content per se. 3) The population view, factor 1 presented by the authors supports the two comments 1 and 2, Nevertheless, thank you note to recognize the altruistic act of trial participation, although important, is not critical. It would have been significant had the 19 participants concluded that, clinical implications of the results was critical for the participants. It is the real trial experience of the patients that is essential to identify critical factor for trial results dissemination. 4) Individual view "There was a desire for trial participants to be able to compare their own experience of being in a trial, and possible side-effects, to that of everyone else in the trial" there lacks clarity, the authors need to clarify especially in the context of the RECAP participants. 5) Provision of trial results summaries to next of kin following death is an essential component of the trial result dissemination. In addition to the perception of the participants, is there a legal framework that permits/restricts passing on the trial results to the next of kin. This is critical for the end-of-life and oncology trial participants. 6) Table 3 lacks clarity and needs to be represented to reflect the parameters compared with the population score. 7) It is uncertain whether consents were obtained form the participants. In spite of the limitation that regulations may vary elsewhere, and that the authors employed Phase III pragmatic effectiveness trials, the observations and conclusions add to the vacuum in the area of clinical trial results dissemination to participants and the public at large.
--	---

REVIEWER	Darbyshire, Julie University of Oxford, NDCN
REVIEW RETURNED	04-Jan-2022

GENERAL COMMENTS	Overall I found this an interesting approach to the question of providing trial participants with results. The development of clear guidelines for this end of study activity has high value to the research community. I struggled to follow some of the points and the statistical summary may need a more expert check. More explanation on the varimax rotation analysis (in particular) may be helpful. I would like to see the data from references 4 & 5 included in the introductory section to quantify how often intention to share results with participants fails to be delivered. This is a key factor guiding the need to provide a template and would save the reader having to find the information from the reference list. For me, table 3 wasn't clear. I'm not sure how the percentages/numericals in the lower half of tables A & B should be interpreted. More explanatory text in the main body of the article would resolve this. Some of the last section of the results would seem to be more appropriately placed in the discussion. There is a mix of reporting findings from the focus group and interpretation of what these mean.
--

	The discussion was rather short. More reflection on the variance of stakeholder involved, and the interactions (including agreements/disagreements) between the types of stakeholder would introduce a socio-cultural view that is currently missing. This would also allow some comment on why/how different types of stakeholder have different priorities. This is mentioned to some extent but more could be made of this, especially patient participants who were looking for personal results rather than overall summaries. I wasn't sure what was meant by this as I would expect most trial participants to have a good idea of their own response to treatment well before the end of trial results? All the commonly recommended points appear to be mentioned in the manuscript but there was no reporting checklist provided. SWATs (page 18, line 464) needs to be in full as well as (or instead of) the acronym.
--	--

VERSION 1 – AUTHOR RESPONSE

Reviewer #1		
4	The Declaration of Helsinki has laid down ethical responsibilities related to clinical trials, and these are restated by the clinical trial regulations in the US, UK, Asia, and the European Union. The guidelines mandate summary results to be provided to study participants in understandable language. Organizing and disseminating trial results to participants is an ethical responsibility. In addition, to the regulatory framework compliance, dissemination has other benefits, aiding in enhancing public trust in clinical research and promoting active participation in future studies. Policy makers and regulators are benefited by easy assimilation of study results in order to generate policies and monitor practices. Regardless of the regulations, three of the key deterrents for the dissemination of trial results have been the concerns of raising public anxiety, and subsequently the fear of legal implications, and loss of public trust. The content, mode, quality, readability, understandability, and selectivity (whether something applies to a subgroup or the whole) of dissemination of clinical trial results still rest in the grey domain of clarity. This is an important study. Given the complexity, the authors have rightly employed a multi-phase mixed-methods triangulation study design using multiple stake holders. The Q methodology is quite appropriate for the study design.	No response required.
5	The paucity of patient participants from concluded clinical trials is a significant shortcoming of the study design. Irrespective of the feedback of the sponsors, public involvement partners. it is the patient and their family reported outcomes that will enable a superior dissemination design. The authors are urged to explain why patient and their families were not abundantly included in the study design?	We agree that the inclusion of trial participants in the RECAP study was of critical importance and it could be argued that more effort could have been made to include additional participants. We have acknowledged this as a limitation. See lines 487-489. 'Yet more could have been done to include a wider

		range of participants with lived trial experience as participants or family members of those participants' However, we do not believe that all included patients and/or people from the publics group would require existing trial experience to contribute in meaningful ways. Most people who take part in clinical trials have not previously taken part in a clinical and as such are not aware of what information they might want to know about the trial at the end. As such we tried to include a range of people (including PPI partners) with and without trial experience. Throughout all stages of the research the patient/public group (which again includes PPI partners) was also the majority group so as to ensure their views were central. With regard to inclusion of family members of people who had participated in trial, whilst this wasn't an explicit inclusion criteria people did share these experiences in the research but as we didn't specify this a priori it wasn't collected as baseline information to describe the sample.
6	Q-sort to determine trial stakeholder views on core content of results summaries is another area of concern. The NHS Research, Ethics Committee members, CTU staff, and PPI partners would have appropriate for the design of the structure of the content, rather than the content itself. The importance of having a patient/patient family led Q-sort is advised to design the content per se.	We were not entirely sure whether the reviewer was suggesting that the content of statements used in the Q-sort should have been more patient/patient and family led or that a higher proportion of patients/patients and families should have been included in the sample that completed the Q-sort to help influence recommendations about the content of feedback to trial participants. As response above to comment #5, we ensured that the patient/public group

		was in the majority during the Q-sort to ensure their views were concentrated. We believed it was also important to include a wide range of stakeholders so as to ensure that the practicability and operationalisation of what information is shared was also considered early. The results from the Q-sort highlight the difference in thinking between patient/public and the other stakeholder groups. This in and of itself is an important finding for trial teams designing results summaries as it highlights that what they think patients want can be very different from what the patients actually want. We have now noted this point in the discussion in line 435-439. 'Overall, the Q-methodology study highlights that patients/publics preferences for information contained within trial results summaries are likely to differ from other trial stakeholders, further underpinning the need to involve this audience during the development of the information.'
7	The population view, factor 1 presented by the authors supports the two comments 1 and 2, Nevertheless, thank you note to recognize the altruistic act of trial participation, although important, is not critical. It would have been significant had the 19 participants concluded that, clinical implications of the results was critical for the participants. It is the real trial experience of the patients that is essential to identify critical factor for trial results dissemination.	We believe that the inclusion of the text in the Discussion in response to comment 5 and 6 above also covers the point raised in this comment.
8	Individual view "There was a desire for trial participants to be able to compare their own experience of being in a trial, and possible side-effects, to that of everyone else in the trial" there lacks clarity, the authors need to clarify especially in the context of the RECAP participants.	We agree this sentence was not clear. Now amended. See line 235-236. 'There was a desire for trial participants to be able to compare their own experiences of possible side-effects to that of everyone else in the trial.'

9	Provision of trial results summaries to next of kin following death is an essential component of the trial result dissemination. In addition to the perception of the participants, is there a legal framework that permits/restricts passing on the trial results to the next of kin. This is critical for the end-of-life and oncology trial participants.	As far as we are aware, there is no legal framework that restricts the sharing of results with next of kin following death. However, as per the findings from RECAP, we would propose that next of kin should have the option of receiving the results (as opposed to assuming desire to receive) and ideally that the participant has consented for the sharing of the results with others. As the reviewer highlights, there may be some trial contexts for which this is particularly relevant. We have acknowledged this in the discussion on line 469-479. ‘There are specific trial contexts in which these considerations may be even more important. For example, an interview study of parents of surviving babies in a neonatal critical care trial identified a delicate balance between the emotionally charged content of the information with defining an endpoint to difficult events and acknowledging participants contributions to the research endeavour [18]. This study also highlighted that even in challenging contexts, trial participants and their families may be responsive to receiving the results of the trial they contributed to and trial teams should endeavour to provide that information in emotionally sensitive ways. As per the recommendations developed by RECAP, we would propose that (where appropriate) participants are invited to consent in advance for the sharing of the results with next of kin, and that next of kin should have the option of receiving the results as opposed to assuming they desire to receive them.’
10	Table 3 lacks clarity and needs to be represented to reflect the parameters compared with the population score.	We agree that Table 3 was not clear. We have now

		edited the tables and the associated text to improve understanding. Please see lines 277-292. 'The most and least frequently identified items from the content analysis of existing trial summaries were compared to the items in the Q set. Firstly, comparison was made of the three most and least important items identified in both the individual and population viewpoints against the frequency of reporting in existing trial result summaries (Table 3A). Overall this highlighted that several items deemed unimportant by Q-sort stakeholders are routinely reported in existing results summaries (i.e. Trial identifier and full title, General information about the trial, Thank you message). The 'Clinical implications' item, which was viewed as important in both Q-sort viewpoints was only reported in 43% of trial results summaries analysed. Table 3B presents the comparison of the three most and least frequently reported items identified in the content analysis of results summaries compared to the scoring within each viewpoint on the Q-sort. Overall there appeared to be similarities on the frequency of items in existing summaries and their relative importance in the viewpoints. However two items identified as being important by the individual viewpoint, 'Issues that may affect the results of the trial' and 'Individual results' this
--	--	--

		was not the case and they were not identified in any of the summaries analysed. Full results of the comparison of the Q-sort and content analysis can be seen Table 3. ‘
11	It is uncertain whether consents were obtained form the participants.	Written consent was sought for the Q-sort and the focus groups. By virtue of the final workshop being a co-production activity (where members are involved in coproducing research are colleagues rather than participants) it did not require informed consent. Good research practice was followed during the activity with appropriate information and support provided to contributors. We have now made this more explicit within the manuscript on lines 560-564. ‘Written consent was sought from participants to publish the data collected (including anonymised quotes) for the Q-sort and focus groups. . By virtue of the final workshop being a co-production activity (where members are involved in coproducing research are colleagues rather than participants) it did not require informed consent. Good research practice was followed during the activity with appropriate information and support provided to contributors.’
12	In spite of the limitation that regulations may vary elsewhere, and that the authors employed Phase III pragmatic effectiveness trials, the observations and conclusions add to the vacuum in the area of clinical trial results dissemination to participants and the public at large.	Thank you.
Reviewer #2		
13	Overall I found this an interesting approach to the question of providing trial participants with results. The development of	Thank you.

	clear guidelines for this end of study activity has high value to the research community.	
14	I struggled to follow some of the points and the statistical summary may need a more expert check. More explanation on the varimax rotation analysis (in particular) may be helpful.	We have now included text within the Q-methodology analysis section of the methods to provide further explanation of Factor analysis and varimax rotation. See lines 135-139. ‘Principal components analysis (PCA), most commonly used in Factor Analysis, with varimax rotation (a statistical technique used at one level of a factor analysis as a way to explain the relationship among factors) was applied to identify relationships between individual Q-sorts. In Factor Analysis, factors are rotated in order to facilitate a more reliable interpretation.’
15	I would like to see the data from references 4 & 5 included in the introductory section to quantify how often intention to share results with participants fails to be delivered. This is a key factor guiding the need to provide a template and would save the reader having to find the information from the reference list.	We have now included text within the introduction to provide further information on trial teams intentions on sharing results. See line 70-75. ‘A recent survey of authors of trials indexed in PubMed identified that only 27% reported having disseminated results to participants with a further 13% planning on doing so, however, 33% had no intention of doing so and the intentions of the remaining 25% was unclear [5]. Also, the reporting of whether and how trial results have been shared with participants was also not done routinely with 74.9% of final reports not mentioning whether results had been shared with participants [4].’
16	For me, table 3 wasn't clear. I'm not sure how the percentages/numericals in the lower half of tables A & B should be interpreted. More explanatory text in the main body of the article would resolve this.	Please see response to Reviewer 1 comment 10.
17	Some of the last section of the results would seem to be more appropriately placed in the discussion. There is a mix of	This section has now been edited throughout to make it

	reporting findings from the focus group and interpretation of what these mean.	clearer that the information presented are findings that were generated by the co-production group and developed into the RECAP recommendations. See lines 390-411. 'It is important to recognise that this was not produced as a prescriptive list but rather suggestions based on diverse stakeholders considered reflections on experiences to date. Broader good practice recommendations to support appropriate reporting of clinical trials results summaries to participants, including who should be involved, were also developed (see Box 1). Overall, these co-produced recommendations highlighted stakeholders views for the need for early planning when considering sharing trial results summaries with participants such that activity is considered from the start and throughout the trial. It was agreed that this should include consideration of who the recipients are (i.e. adults, children, next of kin) which may in turn require different versions. The group also identified the Importance of considering who else needs to be involved in the process e.g. patients organisations, clinicians, etc and ensure they are also committed to the activity. Discussion on the mode of delivery for the result summaries concluded it should be decided in collaboration with trial participants and ideally based on personal preferences. Whether and
--	---	---

		how results summaries will be shared with trial participants who withdraw from the trial was also identified as important by the group as requiring attention. Stakeholders stated that trial teams should also consider when results will be shared and make this clear to participants to set expectations accordingly. If results are not to be shared (due to politically/culturally sensitive issues), this should also be made clear to participants from the outset. When considering who beyond the trial team, the co-production group highlighted that funders should provide financial resource and ask for its inclusion in grant applications, regulatory bodies could provide a repository of good practice to provide exemplars, and employers should source training on writing lay summaries. ‘
18	The discussion was rather short. More reflection on the variance of stakeholder involved, and the interactions (including agreements/disagreements) between the types of stakeholder would introduce a socio-cultural view that is currently missing. This would also allow some comment on why/how different types of stakeholder have different priorities. This is mentioned to some extent but more could be made of this, especially patient participants who were looking for personal results rather than overall summaries. I wasn't sure what was meant by this as I would expect most trial participants to have a good idea of their own response to treatment well before the end of trial results?	We have now extended several sections within the discussion to elaborate on existing points but also included a new paragraph to cover considerations related to how socio-cultural influences may impact on the preferences and needs of participants when sharing results. See Discussion section pages 19-20. Lines 435-439 ‘Overall, the Q-methodology study highlights that patients/publics preferences for information contained within trial results summaries are likely to differ from other trial stakeholders, further underpinning the need to

		involve patients/public during the development of the information. Lines 441-450. A further consideration about developing trial results summaries relates to socio-cultural considerations and how they influence the desire for and presentation of trial result summaries. With regard to content, for some trials and populations there may be a requirement to consider whether some results are sensitive to characteristics such as sexuality or religion and ensure these are accounted for during development. When considering how to share results, it may be more or less appropriate to provide results online for some communities, where in others it may be important to consider who is delivering the message (e.g. a church or community leader may be more appropriate than the trial team). Further research to explore the most appropriate ways to consider the intersections between these characteristics is needed given most research to date has not considered a wide range of diverse experiences [ref Bruhn].’ Lines 458-463 ‘Trial teams could share examples of how trial results have been communicated with trial participants to help inform others teams decision making. A recent good practice example of this involved sharing results of a
--	--	---

		trial through an online meeting (enforced by the pandemic but allowing a more responsive way to share results). The report of this activity also included the associated costs , which will again help teams plan for such activities in future [17].’ Lines 469-479 ‘There are specific trial contexts in which these considerations may be even more important. For example, an interview study of parents of surviving babies in a neonatal critical care trial identified a delicate balance between the emotionally charged content of the information with defining an endpoint to difficult events and acknowledging participants contributions to the research endeavour [18]. This study also highlighted that even in challenging contexts, trial participants and their families may be responsive to receiving the results of the trial they contributed to and trial teams should endeavour to provide that information in emotionally sensitive ways. As per the recommendations developed by RECAP, we would propose that (where appropriate) participants are invited to consent in advance for the sharing of the results with next of kin, and that next of kin should have the option of receiving the results as opposed to assuming they desire to receive them.’
19	All the commonly recommended points appear to be mentioned in the manuscript but there was no reporting checklist provided.	As far as we are aware there isn't a relevant reporting checklist for a multi-phase study of this type.
20	SWATs (page 18, line 464) needs to be in full as well as (or instead of) the acronym.	Now edited. See line 521.

		'(Studies Within A Trial (SWATs))'
--	--	------------------------------------

VERSION 2 – REVIEW

REVIEWER	Nair, Satish Tawam Hospital, Acad affairs
REVIEW RETURNED	14-Feb-2022

GENERAL COMMENTS	The authors have responded to the reviewer queries. No further concerns or questions. The manuscript meets the quality standards of the BMJ open for acceptance.
--

REVIEWER	Darbyshire, Julie University of Oxford, NDCN
REVIEW RETURNED	16-Feb-2022

GENERAL COMMENTS	The tracked change doc supplied seems to indicate significant reviewer comments have been addressed although a point by point response doc would have made this easier to assess. The manuscript is clearer now as a result of the additional information included. I have no further suggestions.
--

VERSION 2 – AUTHOR RESPONSE

We have now edited the 'Strengths and limitations' section of the manuscript (after the abstract).